

# African savanna elephants (*Loxodonta africana*) as an example of a herbivore making movement choices based on nutritional needs

Fiona Sach[1,2], Ellen S. Dierenfeld[3,4], Simon C. Langley-Evans[1,2], Michael J. Watts[1] and Lisa Yon[1,5]

[1] Inorganic Geochemistry, Centre for Environmental Geochemistry, British Geological Survey, Nottingham, UK
[2] School of Biosciences, University of Nottingham, Sutton Bonington, UK
[3] Ellen Dierenfeld LLC, Saint Louis, MO, USA
[4] School of Animal, Rural & Environmental Sciences, Nottingham Trent University, Nottingham, UK
[5] School of Veterinary Medicine and Science, Faculty of Medical & Health Sciences, The University of Nottingham, Sutton Bonington, United Kingdom

Corresponding author
Lisa Yon, lisa.yon@nottingham.ac.uk

## ABSTRACT

**Background:** The increasing human population and global intensification of agriculture have had a major impact on the world's natural ecosystems and caused devastating effects on populations of mega-herbivores such as the African savanna elephants, through habitat reduction and fragmentation and increased human–animal conflict. Animals with vast home ranges are forced into increasingly smaller geographical areas, often restricted by fencing or encroaching anthropogenic activities, resulting in huge pressures on these areas to meet the animals' resource needs. This can present a nutritional challenge and cause animals to adapt their movement patterns to meet their dietary needs for specific minerals, potentially causing human–animal conflict. The aim of this review is to consolidate understanding of nutritional drivers for animal movement, especially that of African savanna elephants and focus the direction of future research. Peer reviewed literature available was generally geographically specific and studies conducted on isolated populations of individual species. African savanna elephants have the capacity to extensively alter the landscape and have been more greatly studied than other herbivores, making them a good example species to use for this review. Alongside this, their movement choices, potentially linked with nutritional drivers could be applicable to a range of other species. Relevant case study examples of other herbivores moving based on nutritional needs are discussed.
**Methods:** Three databases were searched in this review: Scopus, Web of Science and Google Scholar, using identified search terms. Inclusion and exclusion criteria were determined and applied as required. Additional grey literature was reviewed as appropriate.
**Results:** Initial searches yielded 1,870 records prior to application of inclusion and exclusion criteria. A less detailed review of grey literature, and additional peer-reviewed literature which did not meet the inclusion criteria but was deemed relevant by the authors was also conducted to ensure thorough coverage of the subject.

**Discussion:** A review of peer reviewed literature was undertaken to examine nutritional drivers for African elephant movement, exploring documented examples from free-ranging African savanna elephants and, where relevant, other herbivore species. This could help inform prediction or mitigation of human–elephant conflict, potentially when animals move according to nutritional needs, and related drivers for this movement. In addition, appropriate grey literature was included to capture current research.

## INTRODUCTION

The African savanna elephant (*Loxodonta africana*) is categorised as vulnerable on the IUCN Red List and free-ranging populations have declined rapidly across Africa since 1970, predominantly as a result of increased poaching and competition for resources with an increasing human population (*Blanc, 2008*). This competition arises due to the intersection of human activities with elephants' home ranges, and much research is devoted to investigating the reasons why the animals move repeatedly through areas which lead them into conflict with humans (*Eltringham, 1990*; *Hoare & Du Toit, 1999*; *Hoare, 2000*). The aims of this review are to examine the current knowledge on the mineral requirements of the African savanna elephant, to consolidate the current understanding of nutritional drivers for African savanna elephant movement, to examine how geochemistry may affect herbivore movement and to consider how this knowledge could be applied to predict and mitigate human–elephant conflict (HEC) in the future. African savanna elephants have the capacity to extensively alter the landscape and have been more extensively studied than other herbivores, making them a good example species to use within this review. Where relevant, examples of other herbivore movement (including other elephant species) based on nutritional needs are included.

Due to their vast food consumption and behaviour, African savanna elephants can cause significant damage to crops and vegetation (*Eltringham, 1990*; *Hoare, 2000*) and pose a risk to human life and infrastructure. Continued increase in the global human population, to 9.7 billion by 2050, and the associated intensification of agriculture will have major impacts on the world's natural ecosystems (*Nyhus, 2016*). This, coupled with a predicted reduction of 200–300 million hectares of wildlife habitat worldwide, will aggravate human–animal conflict. Wide ranging landscape-level herbivores are increasingly threatened globally (*Wall et al., 2013*). Habitat encroachment and fragmentation poses a substantial threat to elephant populations, forcing them to condense into ever-smaller geographical areas or fenced reserves, whilst putting increased pressure on these areas to meet the animals' resource needs (*Nyhus, 2016*). This can present a nutritional challenge and might cause animals to adapt their movement patterns

to meet their dietary needs, including for specific minerals, presenting wildlife managers with new management issues.

It is the aim of this review to consolidate understanding of nutritional drivers for animal movement especially those of the African savanna elephant, and focus the direction of future research. This will be achieved with the following objectives:

Examine current knowledge on mineral requirements in elephants, including the differences between the nutritional needs of cows and bulls and the activity budget of the species, including time spent feeding.

Examine the relationship between the geochemistry and the associated soil of an area, and how this can alter the minerals available in plants to elephants as consumers (herbivores). Use this information to examine how geochemistry may act as a driver for African savanna elephant movement. Only minerals are being considered within this review.

Consider how knowledge of mineral distribution in the landscape could be used to predict and mitigate HEC in the future.

This review is intended to benefit conservation managers, ecologists, conservation biologists, national park management authorities and potentially managers of animals under human care both within zoos and fenced reserves.

## METHODS

The following method was used to ensure comprehensive and unbiased coverage of the literature. Published studies were identified from three databases, using a range of search terms relating to elephant movement choices.

**Search terms:**

List 1: 'elephant', 'Elephantidae', 'Loxodonta', 'mega herbivore'
List 2: 'soil', 'mineral', 'minerals', 'nutrition' 'geochemistry' 'movement'

The clause 'and' was included between each word in list 1 and list 2. Each search contained 1 word from list 1 and one from list 2. Each word from each list was searched together.

Search terms were selected based on a scan of the literature to give broad covering of subject of interest.

**Databases searched:** Scopus, Web of Science and Google Scholar (searched up to 1st April 2018).

**Fields searched:** titles, keywords, abstracts

**Inclusion/exclusion criteria:**

Only publications which met the following criteria were included in this review. The publication:

Contained at least one of the search terms from each list in the abstract, title or keywords.

Was in a published peer-reviewed journal.

Was in English.
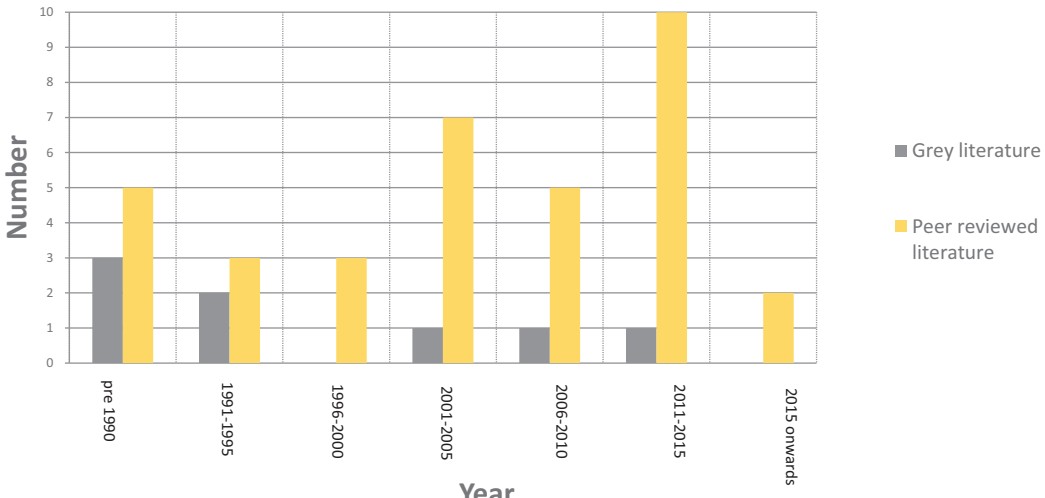

**Figure 1** **Breakdown of the literature by date after the application of the inclusion/exclusion criteria.**

Was relevant to the subject matter (e.g. excluded irrelevant terms such as elephant grass *Pennisetum purpureum*).

**Grey literature reviewed:**

Additionally, relevant grey literature which did not meet the inclusion criteria was reviewed. This was identified as follows:

1. Theses, books and conference proceedings
2. Using internet searches of key terms and snowballing by searching the reference lists of relevant literature (*Sayers, 2007*). Keywords were selected based on a scan of the literature, to give a broad coverage of the subject of interest.

# RESULTS

Initial searches yielded 1,870 records. After applying the inclusion/exclusion criteria, 35 papers were fully reviewed, detailed in Appendix 1. Current work was generally geographically specific and conducted on isolated populations of individual species with dates ranging from 1969 to 2018. Further details of the breakdown of the literature search can be seen in Fig. 1. All reviewed papers were on free-ranging African savanna elephants or other herbivore species including wildebeest (*Connochaetes taurinus*), zebra (*Equus burchelli*), roe deer (*Capreolus capreolus*) and black rhino (*Diceros bicornis*). A total of 70% of reviewed papers focused specifically on African savanna elephants, 30% of reviewed papers focused more broadly on herbivores.

From the review of the grey literature, and additional peer-reviewed literature that did not meet the inclusion criteria, eight further references were identified, which consisted of five books, one thesis and one short report, as detailed in Fig. 1. Dates of references ranged from 1977 to 2012, detailed in Appendix 1.

# ELEPHANT NUTRITIONAL NEEDS

## Challenges of estimating elephant nutritional requirements

Due to the lack of knowledge on the digestive physiology of many wild animals, animal nutritionists use domestic species as physiologic models when designing diets for captive exotic animals. For large hindgut fermenters like elephants and rhinos, the recommendations given for the domestic horse (*National Research Council, 1989*) have been suggested as the appropriate model for most nutrients, due to the similarities in gastrointestinal tract anatomy (*Clauss, Kienzle & Wiesner, 2003*). Therefore, when assessing published nutritional recommendations, the benefits and limitations of using this model must be considered. This approach of using the domestic horse model was validated for white rhinos (*Ceratotherium simum*) and Indian rhinos (*Rhinoceros unicornis*) but not for black rhinos (*D. bicornis*) or any elephant species (*Clauss, Kienzle & Wiesner, 2003*). *Clauss et al. (2007)* demonstrated that black rhinos absorb micronutrients in the same manner as horses, and suggested the same may apply in elephant species. Despite the lack of validation, the horse has been extensively used as a model for captive elephant nutritional physiology (*Olson, 2004*; *Clauss et al., 2007*; *Walter, 2010*) and overall, it is considered a suitable model for many aspects of elephant digestion including the mechanisms by which dietary supplements and dietary crude fibre content influence digestibility, calcium absorption and faecal volatile fatty acid composition. However, elephants have a faster ingesta passage rate than horses, with a total gut transit time of 11–46 h, compared to an average of 48 h in horses, and thus digestibility coefficients are lower for all nutrients (*Bax & Sheldrick, 1963*; *Clauss, Kienzle & Wiesner, 2003*). This must be factored into any comparison with domestic horse recommendations and extrapolation be used with caution.

## Reported mineral deficiencies in captive and free-ranging elephants

As the evidence for specific mineral needs for elephants (of either species) is very limited, documented values for requirements of both African and Asian elephants (*Elephas maximus*) are included for four key minerals; calcium, iodine, iron and zinc. Recommendations are often made on a dry matter basis (DM; feed excluding moisture content). Because Asian elephants are held in greater numbers in captivity, there has been more research on the mineral needs of this species.

### Calcium

It has been suggested that elephants have their highest calcium demands when lactating (females) followed by during periods of intensive tusk growth (*Dierenfeld, 2008*). Calcium metabolism in elephants appears to be similar to that of domestic horses, with approximately 60% absorbed from the diet directly in the intestines, independent of total consumption or requirement, with excess excreted in the urine (*Ullrey, Crissey & Hintz, 1997*). As with other mammals, elephants maintain serum calcium within a narrow range through intestinal absorption, renal excretion and mobilisation of bone (*Ullrey, Crissey & Hintz, 1997*; *Clauss, Kienzle & Wiesner, 2003*).

    *Partington (2012)*, while assessing calcium intake in elephants at 14 UK zoos, determined that a minimum of 0.33–0.77% DM calcium was provided in the offered diets

(values represented minimums as calcium provision from grass or browse forages was not included in the calculations). Nonetheless, even the minimum calculated concentrations exceeded the captive adult elephant maintenance recommendation of 0.3% DM calcium (*Ullrey, Crissey & Hintz, 1997*). Similarly, diets fed to zoo elephants in seven elephant-holding Brazilian zoos contained on average 0.7% DM calcium, showing that minimum recommended levels were being met (*Carneiro et al., 2015*). Diets of semi-captive Asian elephants in India contained 0.46–0.58% DM calcium (*Das et al., 2015*) further supporting the conclusion that calcium deficiencies have rarely been documented in healthy adult captive elephants on maintenance diets. There is, however, evidence that incidence of calcium deficiency is higher in cows during partition and lactation, when calcium demand is increased (*Van Der Kolk et al., 2008*). Subclinical hypocalcaemia was reported in Asian elephants immediately prior to partition at Rotterdam Zoo when calcium demand was not met through dietary provision (*Van Der Kolk et al., 2008*).

Metabolic bone disease (rickets) was reported in captive hand-reared Asian elephant calves. This disease results from an imbalance in the calcium to phosphorus ratio or from intestinal malabsorption, and unbalanced milk formulation may have played a role in this case (*Ensley et al., 1994*).

### Iodine

The thyroid mass of an elephant relative to its body mass is double its predicted size, compared to other mammals (*Milewski, 2000*). This may indicate that the iodine requirements of elephants are proportionally higher than those of other herbivores, and that due to the exclusively herbivorous diet of elephants, they may be susceptible to iodine deficiency (*Milewski, 2000*). Due to the lack of essentiality of iodine to plant metabolism, land plants have little reason to translocate iodine from soil to foliage, therefore plants consumed by elephants may be low to deficient in iodine (*Shetaya et al., 2012*; *Humphrey et al., 2018*). Soil dust deposition has been documented to increase iodine levels of foliage in some situations (*Watts et al., 2015*). As an alternative iodine source, elephants may seek iodine supplementation from iodine rich water or soil (via geophagy). Humans in Malawi were able to obtain as much as 70% of daily iodine requirements from drinking two L of borehole water per day (*Watts et al., 2015*). Iodine is required for reproduction, and the high reproductive success of elephants in conservation areas such as Addo Elephant Park, which contained several boreholes, was hypothesised to be linked with an increased supply of iodine (*Milewski, 2000*; *Milewski & Dierenfeld, 2012*).

In the Kitum caves, Mount Elgon, Kenya, elephants consume the cave salts that correlate with high levels of calcium, sodium, magnesium and phosphorus (*Bowell, Warren & Redmond, 1996*). Iodine was measured in the salt crusts at 1,149 mg/kg, which was >100 times higher than iodine concentrations in the most iodine-rich soils in the vicinity. Reproductive outputs of elephant populations consuming these minerals were also high (*Bowell, Warren & Redmond, 1996*). Given these various lines of inferential evidence, supply or restriction of iodine-rich bore holes could be further investigated as an effective method of population control in situ, without affecting reproductive success

of smaller herbivores that may have a proportionally lower requirements for iodine, which could be realised by diet, water or geophagy (*Milewski, 2000*; *Milewski & Dierenfeld, 2012*).

### Iron

Iron deficiency anaemia has rarely been reported in captive or free-ranging elephants, although several cases of anaemia linked with liver fluke infection, retained placenta, tuberculosis, tuberculosis treatment and malabsorption syndrome have been documented (*Dierenfeld, 2008*). Only a single reported iron deficiency anaemia related to low dietary iron intake, affecting three newly imported Asian elephants, was documented. In this case, clinical signs resolved upon dietary supplementation (*Kuntze & Hunsdorff, 1978*). Diets of semi-captive Asian elephants contained 105–126 mg iron/kg DM (*Das et al., 2015*), considerably in excess of the Nutrition Advisory group recommendation of 50 mg iron/kg DM (*Ullrey, Crissey & Hintz, 1997*; *Das et al., 2015*).

### Zinc

The dietary recommendation for zinc in captive elephants is 40 mg/kg DM diet, based on determined requirements of domestic horses (*Olson, 2004*; *Ullrey, Crissey & Hintz, 1997*). *Partington (2012)* reported zinc levels of between 22 and 52 mg/kg DM in zoo elephant diets offered in 14 UK facilities. However, this figure does not account for zinc provision from grass and/or browse forages, which comprise the majority of the diets, hence these data are limited. Nonetheless the lower end values suggest that some animals may have been consuming inadequate levels of dietary zinc. Semi-captive Asian elephants in India were reported to consume diets containing levels of zinc between 38.4 to 45.9 mg/kg DM (*Das et al., 2015*); no clinical signs of deficiency were seen and serum concentrations were within the ranges reported for healthy elephants (*Ullrey, Crissey & Hintz, 1997*; *Das et al., 2015*). Excess dietary calcium was observed to interfere with zinc bioavailability resulting in skin abnormalities in a zoo elephant (*Schmidt, 1989*; *Dierenfeld, 2008*). *Schmidt (1989)* reported a case of zinc deficiency in a captive Asian elephant, resulting in secondary immune deficiency and skin lesions. The dietary zinc level in that individual was increased from 22 to 54 mg/kg DM and significant clinical improvement was seen within 2 weeks, with lesions resolved after 8 weeks.

Together, these observations confirm that the domestic horse may indeed provide a suitable physiologic model for mineral nutrition of elephants.

## African savanna elephant feeding behaviour

African savanna elephants (*L. africana*) consume a variety of plant material including grasses, leaves, twigs, fruits, barks, herbaceous material and soil (*Kabigumila, 1993*; *Dierenfeld, 2008*). Although described as generalist herbivores consuming over 400 species of plants, diet composition may vary regionally and seasonally (*Kabigumila, 1993*). African savanna elephants are predominantly seasonal grazers and browsers with fruit, barks and soil being consumed as secondary food choices (*Kabigumila, 1993*). There is debate as to whether savanna elephants are predominantly grazers or browsers, with evidence supporting both feeding strategies: *Williamson (1975)* reported elephant diets in

Hwange National Park, Zimbabwe to consist almost entirely of woody plants whereas *Wing & Buss (1970)* reported that elephants in Uganda relied primarily on grasses (approximately 90% of bulk) and therefore labelled the species as grazers. Such geographical variations in diet have prompted some authors to classify elephants as browsers (*Jachmann & Bell, 1985*), whereas others maintain they are primarily grazers (*Beekman & Prins, 1989*; *Tangley, 1997*). Therefore, it is thought that savanna elephants adopt both feeding strategies, and switch depending on environment and season.

Several studies indicate that savanna elephants spend over half of their daily time budget feeding. Elephants in Tsavo National Park, Kenya were observed to feed for 48–63% of daylight hours (*Dougall & Sheldrick, 1964*) and elephants in Lake Manyara National Park, Tanzania were observed to spend on average 76% of daylight hours feeding (*Beekman & Prins, 1989*). Where feeding conditions improved and food availability increased, *Guy (1975)* observed elephants in Zimbabwe to reduce the total amount of time spent feeding to 50–60% of overall time budget, from a greater proportion of their time budget when food resources were limited. Likewise, savanna elephants in areas of food scarcity in Uganda were reported by *Beekman & Prins (1989)* to spend as much as 74% of their total time budget feeding. Flexibility in food items consumed and time spent feeding, indicated that elephants respond and adapt their feeding strategy according to varying availability of food resources.

Savanna elephants have been documented to feed throughout the day, with decreased feeding and increased resting during the middle part of the day; 12:00–14:00 h (*Laws, 1970*; *Beekman & Prins, 1989*; *Shannon et al., 2008*). This pattern was observed in both sexes. Seasonally, the total amount of time spent feeding per day has not been documented to change, although elephants were observed by *Shannon et al. (2008)* to adjust the time of day spent feeding in the hotter summer months. Evidence suggests that plant selection and feeding strategy changes depending upon availability. During the wet season elephants were observed by *Beekman & Prins (1989)* to spend 67% of time grazing with 8% browsing, whilst during the dry season proportions shifted to 23% of time grazing and 60% browsing. During the dry season, the protein content of the grasses decreased. When the protein content of the grasses dropped to <2.5%, elephants in Tanzania were seen by *Barnes (1982)* to increase their browse consumption. Browse typically contains higher levels of secondary compounds such as tannins than grass (*Ellis, 1990*) and thus, as a by-product of this intensified browse consumption during the dry season, tannin and associated levels of toxin accumulation were seen to increase (*Barnes, 1982*).

Mineral levels in plants vary seasonally, geographically and between different parts of the plant (*Joy et al., 2015*) (Table 1 provides specific examples). Due to the generalist feeding nature of African savanna elephants, it is thought that they are able to adapt their food selection as required to meet their target levels of (as yet undetermined) mineral requirements (*Bax & Sheldrick, 1963*). This was demonstrated in elephants within the Kruger National Park (KNP), South Africa, where there is substantial geographical and seasonal variation in plant types consumed by elephants (*Codron et al., 2006*). Stable carbon isotope analysis of faecal material indicated that during the dry season, elephants in northern KNP consumed significantly more grass than their southern counterparts;

**Table 1 Macro-mineral concentrations (% dry matter) in native plants consumed by African elephants (*Loxodonta africana*) in southern and eastern Africa.**

| Location | Season | Plant part | Calcium | Phosphorus | Magnesium | Sodium | Source |
|---|---|---|---|---|---|---|---|
| Hwange National Park, Zimbabwe | Unknown | Mature leaves | 0.02–3.12 | | 0.08–0.64 | 0.02–0.06 | *Holdø, Dudley & McDowell (2002)* |
| | | Young leaves | 0.01–1.32 | | 0.1–0.57 | 0.005–0.05 | |
| | | Stems, twigs | 0.11–1.85 | | 0.02–0.20 | 0.001–0.02 | |
| | | Bark | 0.13–3.93 | | 0.01–0.33 | <0.001–0.02 | |
| | End wet season | Browse | 0.35–2.47 | 0.11–0.33 | | | *Williamson (1975)* |
| | | Grass | 0.41–0.66 | 0.09–0.20 | | | |
| Kasungu National Park, Malawi | Unknown | Tree leaves (12 sp.) | | | | 0.10–1.25 | *Jachmann & Bell (1985)* |
| Tsavo National Park, Kenya | Unknown | Grass and browse (59 sp.) | 0.37–3.61 | 0.08–0.36 | | 0.01–1.67 | *Dougall & Sheldrick (1964)* |
| | Wet season | Mixed plant sp. | 0.13–0.38 | | | | *McCullagh (1969)* |
| | Dry season | | 0.38 | | | | |
| | Unknown | Grasses and herb type vegetation | 0.36–1.44 | | | | |
| | | Shrub | 0.53–8.92 | | | | |

40% of their diet was grass in the northern part of the park during the dry season, compared to just 10% in southern KNP (*Codron et al., 2006*). In contrast, this difference in grass consumption between elephants in the northern and southern parts of this national park was not apparent during the wet season, when elephants throughout the park consumed grass at approximately 50% of their overall diet (*Codron et al., 2006*). This is in accordance of the observed trend of increased grass consumption during the wet season (*Beekman & Prins, 1989*). Although elephants consume a vast number of different plant species, they generally receive the bulk of their diet from a few selected species which vary seasonally and geographically (*Meissner et al., 1990*; *Kabigumila, 1993*). *Bax & Sheldrick (1963)* observed elephants in the Tsavo National Park, Kenya, to select specific plant parts, notably bark rich in calcium.

Free-ranging African savannah elephant daily food intake is estimated from either the weight of the stomach contents (post-mortem) or from extrapolation of data on feeding rates and time spent feeding. Both methods have produced similar estimates of daily DM intake for adults of about 1.0–1.5% of body weight (*Meissner et al., 1990*; *De Villiers et al., 1991*; *Ullrey, Crissey & Hintz, 1997*). DM intake relative to body weight is influenced by a number of factors: DM digestibility, environmental stressors, activity levels and life stage of the animal (adult maintenance, growth, pregnancy or lactation) (*Meissner et al., 1990*). *Laws (1970)* concluded that non-pregnant females and males consumed 1.0–1.2% BW DM (percentage of body weight on a DM basis whereas pregnant females consumed 1.2–1.5% BW DM. On an as-fed basis (feed including moisture content) elephants consumed about 4% of their body weight per day (*Laws, 1970*).

Evidence shows differences between elephant bulls and reproductively active cows in their nutritional needs and associated diet choices, with cows possibly requiring higher levels of minerals and protein to support growing calves (*Dierenfeld, 2008*). *Greyling (2004)*

**Table 2 Reported dietary mineral recommendations for African elephants (*Loxodonta africana*).**

| Mineral | Species | Detail | Daily estimated mineral requirements | Source |
|---------|---------|--------|--------------------------------------|--------|
| Calcium | *L. africana* | Lactating females intensive tusk growth | 60 g 8–9 g | *McCullagh (1969)* and *Dierenfeld (2008)* |
| Sodium | *L. africana* | | 9 mg Na kg$^{-1}$ BW | *Holdø, Dudley & McDowell (2002)* |
| Iodine | *L. africana* | | 0.03 mg I kg$^{-1}$ BW | *Milewski (2000)* |

Note:
Estimated mineral requirements for African elephants.

documented that in the Associated Private Nature Reserves (APNR), South Africa, there was a nutritional difference between various parts of the plants consumed by savanna elephants, with leaves containing more calcium and phosphorus than twigs. It is therefore suggested that cows and bulls meet their differing nutritional needs primarily through plant part selection. Family groups with pregnant and lactating females consumed proportionally more leaves and bark in their diet compared to bulls. In the dry season, females consumed 3% leaves and 14% bark, whereas males consumed 1% leaves and 6% bark and additional twigs (*Greyling, 2004*). This agreed with the previous work of *Stokke & Du Toit (2002)*, who found that bulls consumed more twigs than cows, and cows engaged in more leaf stripping than bulls.

*Greyling (2004)* also documented bulls to consume more plant species with a higher calcium content than adult cows at maintenance (without calves) throughout the year. Greyling suggested that such mineral selectivity may be due to a higher calcium requirement for tusk growth in males compared to females at maintenance. This observation supports previous work conducted by *McCullagh (1969)* who suggested a calcium requirement for male elephants of 8–9 g/day. Additionally, lactating females were found to have significantly higher calcium needs than adult females at maintenance, as summarised in Table 2.

During the dry season, *Greyling (2004)* found bull faeces contained significantly lower phosphorus levels than faeces of cows in family groups. On average, cow faecal samples contained 18% more phosphorus than bulls. Faecal phosphorus levels have been used in agriculture to estimate dietary phosphorus in livestock, and they are a more reliable index to diet quality than faecal nitrogen as they are not influenced by tannins (*Holechek et al., 1985*; *Wu, Satter & Sojo, 2000*). Lower faecal phosphorus in bulls suggests that less phosphorus was consumed in the diet, which might indicate that the requirement for bulls was lower than that of cows (*Grant, Meissner & Schultheiss, 1995*; *Wrench, Meissner & Grant, 1997*).

Feeding time budgets of populations of both sexes, studied in three reserves in South Africa, were found to be similar (*Shannon et al., 2008*). This suggests that cows obtained the required increased dietary energy for pregnancy or lactation, by altering plant selection to preferentially select more energy dense plants, rather than by increasing time spent feeding (*Shannon et al., 2008*). This finding contradicts that of *Guy (1975)*,

who concluded that bulls consumed more 'trunk fulls' of plant material per minute than cows, especially during the dry season, and bulls stayed for longer at feeding sites than family groups (*Stokke & Du Toit, 2002*). Stomach fill post mortem of non-pregnant or lactating females and males was smaller than that of pregnant and lactating females, suggesting that females increased their overall food consumption to meet the nutritional demands of pregnancy and lactation (*Laws, 1970*). These pieces of mixed evidence suggest that several feeding strategies may be adopted by elephant cows and bulls to meet their specific individual nutritional needs, depending upon the unique environments in which they live, and seasonal resources available to them.

Documented literature on specific mineral needs in elephants is very limited and requirements per se have not been experimentally determined (*Das et al., 2015*). Table 2 documents minerals for which estimates have been recorded for African elephants directly. As these values were reached from various different studies, on different populations (captive and free-ranging), parameters of measurement were different, for example, grams required per day compared to mg required per kg DM intake or body weight of the animal. This table does not include requirements extrapolated from domestic horses.

## Elephant movement patters, as related to geochemistry/nutritional factors

The availability of minerals to the plant from the soil underpins the relationship between herbivores and their food supply. The distribution of vegetation was suggested to be strongly associated with the geomorphology of the soil (*Lawson, Jenik & Armstrong-Mensah, 1968*; *Bell, 1982*). Generally, plants will reflect the soil profile and those growing in mineral deficient areas will lack key minerals, thus potentially resulting in deficiencies in the consumer. In contrast, those growing in mineral abundant areas will reflect this, and pass the mineral abundance on to the consumer organism (*Hurst et al., 2013*; *Joy et al., 2015*). The ability of an area to supply minerals to an animal does not solely depend on the mineral status of the soil and geochemical parameters (such as organic matter and soil pH), but also on the ability of the plant to incorporate the minerals (*Bowell & Ansah, 1994*). Additional factors affect the mineral levels within a plant: the pathway of nutrients from the soil to the plant depends upon the amount of element present, the various soil factors that affect the minerals' bioavailability and the plant factors which determine the rate of uptake of the mineral (*Maskall & Thornton, 1996*).

Soil factors which affect a mineral's soil-to-plant transfer include the composition of the parent material, quantity and composition of organic matter and the soil pH (*Hurst et al., 2013*). The relationship between mineral status of the soil and parent rock was strongest where there was minimal chemical weathering (*Bowell & Ansah, 1994*). Organic matter also affects bioavailability, especially that of iodine (*Shetaya et al., 2012*; *Humphrey et al., 2018*). Soil pH greatly influences the metal availability (*Maskall & Thornton, 1996*); in alkali soils, generally the bioavailability of molybdenum and selenium increases, whilst that of copper, cobalt and nickel decreases (*Sutton, Maskall & Thornton, 2002*). Further, increased availability of phosphorus in alkaline soil contributes to its enhanced uptake into the plant (*Maskall & Thornton, 1996*; *Sutton, Maskall & Thornton, 2002*).

Plant factors affecting rate of uptake of a mineral include: age of the plant (with levels of trace elements decreasing in older plants), rate of plant growth (with rapidly growing plants displaying reduced levels of trace elements) and plant species (with differences seen between levels of trace elements in different plant species grown in the same soil (*Maskall & Thornton, 1996*). The greatest differences in mineral content were reported between grasses and browses (*Gomide et al., 1969*; *Ben-Shahar & Coe, 1992*). Seasonally, trace element levels were reported to be higher in plants in the wet season: in the grazing pastures in the Kenyan highlands (*Howard & Burder, 1962*), in grasses by Lake Nakuru in the Rift Valley (*Maskall & Thornton, 1991*) and in the Mole National Park, Ghana (*Bowell & Ansah, 1994*). Finally, grazing status of the plant was seen to influence plant mineral levels, with increased mineral concentrations of up to 300% in grazed areas, notably sodium, phosphorus and calcium, compared to ungrazed areas supporting low animal densities (*McNaughton, 1988*).

Forage mineral analysis data is routinely used to assess mineral levels in agriculture, and despite its limitations, it is believed to be a reliable index to be used to assess the general ability of forages to meet animals' mineral needs (*McNaughton, 1988*; *Nellemann, Moe & Rutina, 2002*). However, the mineral profile of the soil can be depleted by soil, plant, topography and weather factors. In the Sabi Sands Reserve, South Africa, 10 species of grass were analysed and grasses from soils of higher mineral levels accumulated lower mineral concentrations, compared to grasses from soils where the minerals were found in lower levels (*Ben-Shahar & Coe, 1992*). In this case, this was thought to be due to sampled species attributes, and the effect of the local micro-climate on the plants.

### Movement choices of elephants

Several studies concluded that elephant habitat use is not random, but that elephants have specific preferences for various habitats and move to fulfil their resource needs (*Whitehouse & Schoeman, 2003*; *Osborn, 2004*; *Douglas-Hamilton, Krink & Vollrath, 2005*; *Dolmia et al., 2007*; *Thomas, Holland & Minot, 2008*; *Leggett, 2015*). There are a myriad of factors that contribute towards elephants' movement choices including availability of food and water, opportunity for social interaction, human presence and associated activities. Hydrology and topography may also influence animal movement (*Bowell & Ansah, 1994*; *Wall, Douglas-Hamilton & Vollrath, 2006*). Elephants tend to avoid steep slopes due to the increased energy expenditure required to climb them; even minor hills can be considerable energy barriers to an elephant (*Wall, Douglas-Hamilton & Vollrath, 2006*). *De Knegt et al. (2011)* suggested that daily movement of elephants related predominantly to food availability, and movements become extended by the distance traversed to water sources. Elephants in that study area of the KNP, South Africa concentrated their foraging within areas of high forage availability that were closest to water, whilst still being large enough areas to optimise efficiency of movement and foraging.

The significance of the impact of human activity on the natural movements of elephants is rapidly increasing (*Nyhus, 2016*). *Tucker et al. (2018)* concluded that in areas with a high level of human presence, mammal movement decreased by 35–50% across 57 species,

compared with areas of low human presence. Over the last 150 years, expansion of human settlement into elephant habitat, and an increase in elephant killing (from poaching and hunting) has significantly altered elephants' home ranges across continental Africa (*Eltringham, 1990*; *Hoare, 2000*; *Osborn, 2004*; *Nyhus, 2016*). Initially it was thought that a simple linear relationship would exist between rising human and declining elephant densities at a national or subcontinental scale (*Hoare & Du Toit, 1999*). However, *Hoare & Du Toit (1999)* found that in an area of 15,000 km$^2$ in northwest Zimbabwe, the relationship turned out to be more complex. Using data from human populations, and observed elephant densities in the region, the authors determined that there was a threshold beyond which elephant and human coexistence could no longer occur, and elephant populations rapidly declined. This threshold was related to agricultural development, and was reached when land was spatially dominated by agricultural use, and the original woodland (that constituted the elephants' habitat) became sub-dominant.

When analysing elephant movement, water availability must be taken into account; elephants are obligate drinkers (*Wall et al., 2013*). Water availability is considered to affect elephant movement, both on a daily and seasonal basis and may be a greater driver for elephant movement than mineral availability. Three studies conducted in South Africa and Kenya indicated that elephant movement increased throughout the wet season when water availability was greatest, and then rapidly decreased throughout the dry season, with elephants, especially lactating females, confining themselves to areas within 1–2 days' travel from water to enable them to conserve energy (*Western & Lindsay, 1984*; *Codron et al., 2006*; *Thomas, Holland & Minot, 2008*; *Birkett et al., 2012*).

*Pretorius et al. (2011)* concluded that elephants made movement choices based on nutritional provision in a specific area. Fertiliser was applied to mopane trees (*Colophospermum mopane*) in the APNR, South Africa, in various patches, resulting in an increase in the phosphorus and nitrogen levels in mopane leaves. Elephants consumed more mopane leaves per patch in fertilised patches compared to unfertilised patches, regardless of patch size. Furthermore at a 100 m$^2$ patch size scale, elephants stripped leaves more in fertilised than unfertilised patches, but were more likely to tree kill (through uprooting or breaking main trunks) in unfertilised patches. Therefore, it was suggested that elephants caused more impact to trees of lower value (through tree killing) whilst preserving trees of higher value (fertilised mopane) through coppicing (*Pretorius et al., 2011*).

Secondly *Pretorius et al. (2012)* suggested that phosphorus may be a key driver for elephant movement, with elephants moving throughout the year to maximise intake of this mineral. In this study area in the APNR, there was a suspected local deficiency in phosphorus, potentially explaining why the elephants prioritised obtaining this mineral. During the wet season, when food availability was greatest, nitrogen provision was prioritised, possibly to meet the elephants' needs for growth and reproduction. During the dry season, when food was potentially limited, energy was prioritised by the elephants. This could be because energy costs to obtain food and water during the dry season were often higher as elephants had to travel further, due to a reduced abundance of forage and availability of water (*Pretorius et al., 2012*).

### Nutritional factors affecting elephant movement

Minerals can be provided to elephants from multiple sources, including plants, water or soil (through geophagy). Examples of mineral provision from plants include sodium, calcium, magnesium and phosphorus. Forest elephants (*L. cyclotis*) in the Kibale National Park, Uganda, were reported by *Rode et al. (2006)* to be crop raiding to meet their sodium need. It was reported in the literature that minerals such as copper and sodium, rather than energy and/or protein, were limited in the elephants' wild food plants, and were found in higher concentrations in crops. Often, wild elephant food plants which are high in sodium are also high in secondary compounds (*Rode et al., 2006*), which might inhibit the uptake of essential minerals and increase sodium excretion, and thus may further exacerbate low sodium intake (*Jachmann, 1989*). Crops contained lower levels of secondary compounds compared to wild plants, which allowed the elephants to resolve the complexities of meeting sodium needs without interference from secondary compounds. For example, the highest sodium-concentration wild plant in this study, *Uvariopsis congensis* also contained high levels of secondary compounds, saponin and had a high alkaloid score (*Jachmann, 1989*). *Jachmann (1989)* has also reported examples of elephant populations in the Miombo biome, Africa, making plant choices to create diets that contained high sodium and digestible sugar concentrations, and low concentrations of indigestible fibre and secondary compounds. The elephants especially avoided plants with high phenol and steroidal saponin levels. Additionally in Kibale National Park, seasonal availability of wild food was not correlated to the timing of crop-raiding events (*Chiyo et al., 2005*). This suggests that elephants may be selecting specific food crops due to their nutritional provision, rather than just being attracted to the presence of food crops and increased overall availability of food (*Chiyo et al., 2005*).

Finally, savanna elephants within the Mount Elgon region, Kenya, consumed salt deposits within the Kitum caves, which are rich in a variety of minerals including calcium, sodium, magnesium and phosphorus (*Bowell, Warren & Redmond, 1996*). Cases of uneven tusk wear were noted and presumed to result from the use of tusks to scrape salts from the ceiling and walls (*Bowell, Warren & Redmond, 1996*). The environment within the cave can be warmer at 13.5 °C than surrounding areas where night temperature can drop to 8 °C, and although this could be encouraging the elephants to remain in the area overnight, it was suggested that there exists a nutritional drive causing them to seek out and consume the salt deposits on the rocks (*Bowell, Warren & Redmond, 1996*).

Minerals can also be provided to elephants through drinking water. *Sienne, Buchwald & Wittemyer (2014)* investigated elephant use of bais (natural forest clearings which often have seasonal or year round sources of water present as surface waters) in the central African rainforest and concluded that mineral provision from water is likely to be attracting elephants to specific bais. Mineral concentrations in water from elephant-evacuated pits were higher than in surface water, and thought to be a causative factor behind bai visitation choice. In particular iodine, sodium, sulphur and zinc were elevated, while calcium, magnesium, manganese, iron and tin concentrations were at least ten times higher in elephant-evacuated water compared to surface waters.

*Blake (2002)* observed that elephants congregated around bais during the dry season, correlating with a seasonal peak in mineral levels in pit water, which may be due to the seasonal ebbing of spring water flow. Likewise, savanna elephants in the Hwange National Park, Zimbabwe were recorded by *Weir (1972)* in greater numbers surrounding water sources with higher sodium content. Pans of high sodium water were reported to have three times as many elephants when censured, compared to the lowest sodium areas, indicating that elephants might make movement choices based upon sodium need (*Weir, 1972*).

Finally, geophagy appears to be a normal behaviour of all elephant species in the majority of habitats and is thought to aid elephants in meeting their nutritional (mineral) needs (*Holdø, Dudley & McDowell, 2002*). There is some evidence that elephants also conduct geophagy to support detoxification of unpalatable secondary compounds in their diet (*Mwangi, Milewski & Wahungu, 2004*; *Chandrajith et al., 2009*). In other ungulate species, clay may decrease the harmful effects of secondary plant compounds and intestinal infections (*Klaus & Schmidg, 1998*; *Ayotte et al., 2006*). Soil is never consumed randomly within an elephants' home range, but instead it is consumed from specific spatially circumscribed sites (*Klaus & Schmidg, 1998*). It is thought that elephants principally consume soil(s) at specialised licks to supplement sodium intake, although calcium, magnesium and potassium are also often higher in lick soils compared to the surrounding soils (*Holdø, Dudley & McDowell, 2002*). Additionally, elephants are known to consume soil on termite mounds, although it remains unclear as to the driving mineral(s) behind this behaviour. In contrast to the situation at lick sites, sodium levels do not seem to be persistently higher in termite mounds than surrounding soils (*Holdø & McDowell, 2004*).

A further example of geophagy by elephants was reported by *Mwangi, Milewski & Wahungu (2004)* in the Aberdares National Park, central Kenya, where elephants rely on browse and unripe fruits to make up the majority of their diet due to a limited availability of grasses. Browse, unripe fruits and seeds generally contain more tannins and alkaloids than grasses, suggesting that the elephants in this national park consume more potentially harmful substances compared to elephants that consume higher levels of grasses. As hindgut fermenters, neutralisation of these harmful substances is not possible in the same way as it is for ruminants (where foregut fermentation is used to neutralise these harmful substances). Since the soils consumed also contained higher levels of sodium and iodine than surrounding soils, it is not possible to identify if minerals or clays are the driving force behind this geophagic behaviour, however, it was considered that both factors were important (*Mwangi, Milewski & Wahungu, 2004*).

In the Kalahari-sand region of Hwange National Park, elephants consumed high-sodium lick soils during the dry season possibly in response to an unmet requirement for sodium (*Holdø, Dudley & McDowell, 2002*). Lactating and pregnant females consumed more soil per visit to a high sodium lick than males (*Holdø, Dudley & McDowell, 2002*). The latter might be due to their increased requirement for sodium during pregnancy and lactation (*Michell, 1995*). This suggests that there is a physiological cause for this geophagy and that in these cases, lick use is driven by a nutritional need. Female elephants

will increase geophagy to meet their additional nutritional needs during pregnancy and lactation (*Michell, 1995*). Table 1 documents sodium levels in browse species during the dry season that are lower than during the wet season, and were suggested by *Holdø, Dudley & McDowell (2002)* to be insufficient. The soil in the mineral lick areas also contained elevated levels of magnesium and calcium, however, these minerals were also available in adequate amounts from other sources such as termite mounds or dietary browse. Interestingly, consumptions of termite mound soils was not observed. Therefore, the authors concluded that these elephants were conducting geophagy based on sodium need (*Holdø, Dudley & McDowell, 2002*).

As well as the increased clay in the soil in the Aberdares National Park, *Mwangi, Milewski & Wahungu (2004)* found that the soil consumed by the elephants also contained higher levels of sodium and iodine than the surrounding areas, but was significantly lower in zinc, manganese and iron levels. Additionally, there was 250% more phosphorus and 50% more magnesium in the consumed soil than the surrounding control soil (*Mwangi, Milewski & Wahungu, 2004*). This suggests that elephants of this population chose to consume soil in certain areas based on nutrition provision, and that specific minerals were prioritised.

There is debate as to whether elephants alter their movements to seek out and consume either the soil from termite mounds, or plant material growing on the termite mounds, to meet their mineral needs (*Holdø & McDowell, 2004*; *Muvengwi, Mbiba & Nyenda, 2013*; *Muvengwi et al., 2014*). Soil from termite mounds includes both surface soil and deeper subsoil, raised to the surface by termites. Previous studies generally focused on one geographical area and thus results may be geographically specific depending upon surrounding mineral availability. It appears to be universally acknowledged that soils from termite mounds contain more minerals than surrounding areas as the termites mine deeply into the substrate (*Holdø & McDowell, 2004*; *Muvengwi, Mbiba & Nyenda, 2013*; *Muvengwi et al., 2014*). However, the evidence as to whether elephants move to seek and consume specific soils (and plants) for targeted minerals is variable. *Muvengwi, Mbiba & Nyenda (2013)* showed that tree diversity did not vary significantly on termite mounds or control plots, in Chewore North, Zimbabwe, yet net biomass removal by mega-herbivores was up to five times higher on control plots than termite mounds. Specifically, when measuring consumption of *Colophospermum mopane*, there was no difference in biomass removal between termite mounds and control plots (*Muvengwi et al., 2014*).

In contrast, black rhino in Chipinge Safari, Zimbabwe, were observed to browse on foliage growing on termite mounds more than off termite mounds, seen by increased bite intensity on the plants from the termite mounds (*Muvengwi et al., 2014*). This is suspected to be due to the increased soil and foliar mineral levels. Concentrations of nitrogen, potassium, phosphorus, calcium and sodium were found to be approximately double in the soil and leaves on termite mounds, compared to those off the termite mounds (*Muvengwi et al., 2014*). In the Kalahari Sand Hwange National Park, Zimbabwe elephants consumed soil from the high sodium, sparsely grassed areas on top of the termite mounds if the surrounding soil had a low concentration of sodium, but not if the

surrounding soil areas had comparably higher sodium content (*Weir, 1969*). In western Zimbabwe, 12 paired sample sites were compared. Each site consisted of an area with a termite mound and a corresponding area within woodland, containing no termite mound. *Holdø & McDowell (2004)* concluded that although the soils within the termite mounds contained more of all tested minerals, the plants on the termite mounds contained less sodium than the plants in woodland plots. Elephants fed more intensively from the plants on the termite mounds than within the woodlands indicating that in this situation, the animals were probably seeking other minerals in addition to sodium from the termite mounds (*Holdø & McDowell, 2004*).

Finally, termite mounds which are consumed by elephants within the Mimbo ecosystem of the Ugalla Game reserve, Tanzania, contained more minerals than termite mounds which are not used for geophagy (*Kalumanga, Mpanduji & Cousins, 2017*). Amounts of each mineral correlated with each other, making it impossible to distinguish a single vs multiple specific driver(s) underlying geophagy. However, it is clear that mineral-rich termite mounds are being selected for consumption over less mineral-rich termite mounds (*Kalumanga, Mpanduji & Cousins, 2017*).

## Applications to ameliorating human–elephant conflict

Human–elephant conflict is caused when elephants make forays into human settlement resulting in some form of damage. Humans retaliate to injure, kill or displace the elephant (*Hoare, 2000*). The African Elephant Specialist Group conducted an inventory of sites across Africa where HEC occurs. It was concluded that the issue is widespread and HEC occurs where interactions happen between the home range of elephants and human activity. Approximately 20% of elephant home range is within legally protected areas, however, conflict was documented to occur in both protected and non-protected areas (*Said et al., 1995*). Crop losses attributed to elephants across Africa was low (5–10%), and elephants were considered to be low on the list of agricultural pests (*Hoare, 2000*; *Naughton-Treves, 2008*). However, widespread low-level damage from non-dangerous crop pests were better tolerated by communities than rare, localised catastrophic damage caused by elephants (*Said et al., 1995*; *Hoare, 2000*; Naughton-Treves, 2008). There is limited evidence to support the relationship between problems caused by elephants and the level of elephant density or nutritional food limitation (*Barnes, Asika & Asamoah-Boateng, 1995*; *Hoare, 1999*). The optimum foraging theory has been suggested to explain the unpredictable nature of crop raiding across the savanna (*Hoare, 1999*). This theory predicts that animals will maximise the quality of nutrient intake where possible and thus when crops of higher nutritional value than wild food plants are available, animals will prioritise consumption over their normal food crops (*Begon, Harper & Townsend, 1986*).

### Applications to other herbivore species in comparable environments

Consideration of geochemistry is required for maintenance of healthy animal populations, especially within fenced reserves where animal migration is impossible. For example, in Lake Nakuru National Park, Kenya which is a fenced area of 160 km$^2$, the soil is derived from volcanic ash, pumice and lake sediment, with low levels of extractable cobalt,

copper and acetic acid with a high alkaline soil pH (*Maskall & Thornton, 1996*). In this region of the Rift Valley, mineral deficiencies including copper and cobalt were seen in domestic cattle, as well as in impala (*Aepyceros melampus*) and waterbuck (*Kobus defass*) (*Maskall & Thornton, 1996*). The increased soil pH caused increased uptake of molybdenum by the plants, which in turn inhibited the utilisation of copper in ruminant animals, further exacerbating the deficiency of copper (*Underwood, 1977*). A geochemical survey was conducted and results of this related to observed clinical copper deficiencies in animals (*Maskall & Thornton, 1996*). Following this investigation, recommendations were made to the Kenya Department of Wildlife Conservation and Management that mineral salts containing cobalt, copper and selenium should be made available to wildlife in the park to mitigate these mineral deficiencies (*Thornton, 2002*). Due to the physiological differences in mechanisms of copper absorption in ruminants and non-ruminants, elephants are not as sensitive to this deficiency as ruminant species and a similar problem has not been documented in elephants (*Maskall & Thornton, 1996*).

Clinically observed copper deficiencies caused by an increased uptake of molybdenum by the plant and thus interference in the utilisation of copper by the animal were seen in Grant's gazelle (*Gazelle granti*) from another area of the Kenyan Rift valley (*Maskall & Thornton, 1996*). Additionally, this was seen in moose (*Alces gigas*) in Alaska (*Kubota, Rieger & Lazar, 1970*) and several herbivores at the San Diego Wild Animal Park (USA) where hypocuprosis was diagnosed, caused by feeding alfalfa with a high molybdenum (and sulphur) concentration (*Kubota, Rieger & Lazar, 1970*; L. Nelson, 1981, unpublished data; *Maskall & Thornton, 1996*). In northeast Zimbabwe, it was suggested that high concentrations of iron in the soil and forage inhibited the availability of phosphorus to the plants, and thus to the cattle consuming the plants. The high iron concentration in the soil also reduced the absorption of copper and zinc in cattle (*Fordyce, Masara & Appleton, 1996*).

Due to the ever-changing environment in which herbivores live, they are forced to make a series of prioritised decisions to ensure survival. These decisions range from spatial to temporal and vary in scale, from smaller scale decisions around which plant part to select for consumption, through to decisions around seasonal movement patterns (*Fryxell, 2008*). *De Knegt et al. (2011)* concluded that forage availability, both in terms of quantity and nutritional quality, varies between seasons and years. Consequently, those individual herbivores adapt their ranging behaviour to meet their nutritional needs and ensure survival. This is especially important in times of resource scarcity, where poor decision making may result in a reduced reproductive output or death (*Shannon et al., 2010*). Appropriate discrimination between food items of high or low quality can thus produce a selective advantage for long term survival (*Fryxell, 2008*).

From tracking data on 803 individuals of 57 species, *Tucker et al. (2018)* concluded that animal movements are on average shorter in resource rich environments. For example, red deer *(Cervus elaphus)* in Slovenia were found to have reduced home ranges due to the enhancement of resources, via supplementary feeding (*Jerina, 2012*), further agreeing with the work conducted by *Morellet et al. (2013)* and *Teitelbaum et al. (2015)*.

*Morellet et al. (2013)* showed that the home range of roe deer *(Capreolus capreolus)* at higher altitudes, was significantly larger than roe deer at lower altitudes, despite forage availability at higher altitudes being more abundant and of higher quality, although the growing season was shorter than at lower altitudes. This suggested that home range, on an individual basis, is linked to a balance between metabolic requirements and ability to acquire food, accounting for seasonal variation. *Teitelbaum et al. (2015)* concluded from a review of 94 land migrations of 25 large herbivore species that there was a 10-fold increase in the migration distance between resource high and low areas. These studies indicated that animals living in resource poor areas will have larger home ranges and longer migration distances than those living in resource abundant areas.

African herbivores are not distributed heterogeneously. In the Serengeti National Park (SNP), areas of high herbivore concentration corresponded with areas providing forages of higher mineral content, implying that mineral content in foods was an important determinant of the spatial distribution of herbivores within this park (*McNaughton, 1988*). For example, magnesium, sodium and phosphorus had a particular influence on herbivore distribution, with high herbivore density areas having 300% more sodium, 50% more phosphorus and 10–23% more magnesium, respectively, than low herbivore density areas. Secondly, migratory grazing ungulate species in the SNP were reported to make seasonal movements based on grass mineral content (*McNaughton, 1990*). Grasses, as is common in many tropical soils, were not sufficient in magnesium and phosphorus to meet the mineral requirements for lactating and growing ruminants, and overall were lower in minerals than grasses growing in temperate soils (*McDowell, 1985*).
The nutritional needs of lactating females and growing young were reported to be influential on movement choices (*McNaughton, 1990*). Animals have evolved with parturition periods being governed by the nutritional requirements of reproducing females and growing young, seasonal rainfall and distance from forage of sufficient quality being prioritised (*McNaughton, 1990*).

Herbivores have responded to plant evolutionary development through exhibiting seasonal habitat selection and a reported change in movement behaviour. This was shown by Shannon et al. (2010), from examining ranging behaviours and broad scale decision making of wildebeest (*Connochaetes taurinus*), Thomson's gazelle (*G. thomsoni thomsoni*), red deer (*Cervus elaphus*), reindeer (*Rangifer tarandus*) and elk (*Cervus canadensis*). Zebra and wildebeest around the Sabi Sands Reserve, South Africa were seen to move seasonally to habitat types characterised by grass communities with a high proportion of nutritious species, and generally increased level of grass diversity, rather than selecting a particularly nutritious species within a broader habitat (*Ben-Shahar & Coe, 1992*). Home range movement showed that diet composition and habitat use of these animals was influenced by the availability of nitrogen and phosphorus in grasses (*Ben-Shahar & Coe, 1992*).

## CONCLUSIONS

Evidence-based values for mineral requirements of elephants remain undetermined. Suspected deficiencies in local key minerals might force animals to make movement

choices to obtain these minerals. In African savanna elephants this behaviour has been reported, although there is a need for further research. The latter might reveal correlation patterns which could aid conservation managers in making informed decisions surrounding elephant movement, and the mitigation of HEC.

This review collates evidence to suggest that African savanna elephants (and other herbivores) consider nutritional drivers as a factor in their movement choices. The reasons dictating an animals' daily, seasonal and annual movement are considered to be multifactorial, with availability of water, human activity, social behaviour and topography all playing a role alongside nutrient availability, specifically mineral provision. Minerals are available to elephants from plants, water and soil and all contribute to meeting their, as yet, undetermined mineral needs. There is a relationship between geochemistry and herbivore movement, respectively mineral provision to the consumer, through consumption of plants, water and soil (through geophagy). This relationship needs to be further explored to aid in predicting animal movement.

National Parks and fenced reserves may occupy marginalised land of poorer quality, which has not been assigned to agriculture. The vast increase in land required from 2014 to 2050 for human population growth and agriculture will lead to a further reduction in land available for herbivores such as savanna elephants, and HEC is predicted to increase (*Nyhus, 2016*). Wide ranging, landscape-level movements made by terrestrial herbivores are increasingly threatened globally (*Wall et al., 2013*). From a practical conservation perspective, there is limited research on the impact mineral provision may have on prediction or mitigation of HEC, and how this could be used as a tool for conflict resolution.

### Funding
This work was supported by the Natural Environment Research Council (grant number NE/L002604/1) and the British Geological Survey University Funding Initiative (BUFI). The funders had no role in study design, data collection and analysis, decision to publish or preparation of the manuscript.

### Grant Disclosures
The following grant information was disclosed by the authors:
Natural Environment Research Council: NE/L002604/1.
British Geological Survey University Funding Initiative (BUFI).

### Competing Interests
Ellen Dierenfeld is employed by Ellen Dierenfeld Consulting, LLC.

### Author Contributions
- Fiona Sach conceived and designed the experiments, performed the experiments, analysed the data, contributed reagents/materials/analysis tools, prepared figures and/or tables, authored or reviewed drafts of the paper, approved the final draft.

- Ellen S. Dierenfeld authored or reviewed drafts of the paper, approved the final draft, extensive review.
- Simon C. Langley-Evans authored or reviewed drafts of the paper, approved the final draft, extensive review.
- Michael J. Watts authored or reviewed drafts of the paper, approved the final draft, extensive review.
- Lisa Yon authored or reviewed drafts of the paper, approved the final draft, extensive review.

## Data Availability

The research in this article did not generate any data or code as it is a literature review paper.

## Supplemental Information

Supplemental information for this article can be found online at http://dx.doi.org/10.7717/peerj.6260#supplemental-information.

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
