# Peer review of "African savanna elephants (Loxodonta africana) as an example of a herbivore making movement choices based on nutritional needs"

_PeerJ, doi:10.7717/peerj.6260_

## Round 0.1 · original submission · Major Revisions

Three reviewers have now provided feedback on your review. While all three commended you for tackling this topic and for providing a novel and important contribution to the field, all three also had some concerns about your review. In general the reviewers request a more detailed description of your methods, an alternative structure to your discussion, and a stronger link between your stated aims and the points raised in your discussion.

I believe that your article will represent a strong contribution to the field, but I cannot accept it for publication with PeerJ in its current form. I will not reiterate the reviewers’ specific comments, but request that you respond to each of their thoughtful and detailed suggestions as you revise your article for resubmission (please note that reviewer 1 has included an annotated manuscript with additional feedback).

Additionally, I have a few points that I wish to note regarding your literature review that I request you respond to:

You need to provide a complete and detailed inventory of all the articles and grey literature that you included in your review, either within your article or as supplementary materials. I encourage you to look at previously-published reviews in PeerJ to guide you e.g.,
Bersacola et al. (2018) DOI: 10.7717/peerj.4847
Goldstein et al. (2018) DOI: 10.7717/peerj.4932
Cronin et al. (2017) DOI: 10.7717/peerj.3649
Rahim & Patton (2015) DOI: 10.7717/peerj737

Further to this point, and as noted by the reviewers, your methods lack detail. For example, how many articles (what %) did you have to exclude because you could not access them in full? Were certain journals omitted entirely from your search because of this limitation? How did you select the specific key words that you chose? Please provide more information regarding this. Again, I suggest you look at previously-published reviews in PeerJ for examples.

Lastly, in your results section you state “A less detailed review of grey literature, and additional peer-reviewed literature which did not meet the inclusion criteria but was deemed relevant by the authors was also conducted to ensure thorough coverage of the subject. Additionally, some key papers on wild elephants were also reviewed.” This sounds like methods more than results. Please provide descriptive statistics about the grey literature included in your results (e.g., number of articles, books etc., topics included etc.) rather than simply how you found them (this information is already provided previously in your methods section and so is redundant).

·

Basic reporting

-The text is written in professional English. Splitting long sentences in further short ones would make it much easier to read and follow the content. Very often “;” is used to link sentences which reduces the text flow. I recommend omitting this wherever possible.

-The introduction shows the background and context of the work sufficiently and clearly. Throughout the text I repeatedly missed citations close to the statements made out of them. The reference list needs thorough revision as indicated in the pdf.

-In my opinion the subject is definitely of broad interest especially regarding the currently existing (and in the future probably aggravating) human-elephant conflict.

-Functioning as basis for a potential approach to mitigate human-elephant (or more broadly human-wildlife) conflict, seems to me a very good reason for this review article.

-The introduction provides sufficient background information and makes clear who the intended audience is.

Experimental design

-The methods are described in a concise manner, providing sufficient detail to replicate the study.

-The chosen method seems appropriate to me to comprehensively coverage the subject. If both the African and the Asian elephant species had been included throughout the study, this might have led to even more complete findings.

-In my opinion, sources are quite often not cited as clear as I prefer it to be (see comments made in the pdf). This might be overcome by citing statements closely and not only at the end of the corresponding section.

Validity of the findings

-Conclusions are stated and linked to the original research questions. Unresolved research questions are identified as well as future fields for research.

Additional comments

-With respect to the situation of free-ranging elephant populations and the increasing conflict with humans, the intention of this article seems very important to me. It combines different fields (elephant nutrition, mineral content in feeding plants, herbivore movement patterns, conservation) of research, which is a quite challenging task.

-In my opinion the structure of the manuscript needs to be revised to make it clearer. I would recommend doing this by the use of the following subheadings:
Author cover page (no adaptation required)
Abstract (no adaptation required)
Introduction (no adaptation required)
Survey methodology (should include text from line 105 to 134)
African elephant feeding behavior
Elephant nutrition requirements and the domestic horse model
Reported mineral requirements and deficiencies in elephants (might be included in the previous subheading)
Effect of geochemistry on elephant dietary intake
Geochemistry influencing animal movement and land use decisions of herbivores
Land use decisions of elephants
Conclusions (no adaptation required)
References

-Considering the human-elephant conflict a keystone of this manuscript, I would encourage to extend the section in the introduction on this subject. In doing so, literature on the situation in Asia where human-elephant conflict is a considerable issue, might be added and improve the background information.

·

Basic reporting

The introduction sets up an interesting research question—the nutritional drivers of elephant movement on human-dominated landscapes—that could be of broad and cross-disciplinary interest. I commend the authors on identifying an important and interesting area for review. However, the paper was unfocused and left me unclear as to how the rest of the paper followed through with what was laid out in the introduction. The stated purpose of the study was to review previous work on elephant movement for mineral needs, and the ways in which elephants are modifying their movement in response to anthropogenic pressures. Crucially, the studies reviewed do not analyze elephant movement or habitat selection for minerals, nor how elephant movement for minerals is influenced by changing human pressures on populations. There were no search terms related to movement or habitat selection. Rather, a range of topics related to elephant nutrition are explored without transitions or previous mention in the introduction (e.g. cow/bull differences, activity budgets), that are better categorized as a discussion of elephant nutrition. This left me returning to the introduction again and again to remind myself what the goal was, and the aims laid out in the discussion conflicted with those in the introduction. I think the manuscript would benefit greatly from a more focused organization and a revision of search terms (there is a substantial body of literature on elephant habitat selection that was excluded).

As this was a review paper, I would like to have seen more detail in the results reporting on the literature search. At present the only detail is the number of articles found before inclusion and exclusion criteria were applied. What was the number after this process was applied? How do those studies break down by year, geographic region, topic, captive vs. wild, etc…? What were the trends found in the literature search regarding the research directions taken? What areas of research have not been addressed in the literature and are deserving of further study?

Without any context for the search results, it is hard to gauge the thoroughness of topics covered in the discussion. Certain sections rely very heavily on old literature, and I am left wondering whether these topics have been abandoned over recent years or whether they have simply been left out of the review for some reason? There are other cases in the manuscript where bold statements are made without any references, and other instances where references are used that do not appear in the reference list (e.g. Sienne et al. 2014).

Professional English was used throughout the manuscript, but the main point of many paragraphs was unclear and transitions between paragraphs lacking. Additionally, there were grammatical errors throughout the manuscript like mismatched subject-verb pairs (e.g. l. 138, ll. 327-328), missing definite articles (l. 74), misplaced/missing punctuation (e.g. l. 390, ll. 467-470, l. 616), incorrect word usage (e.g. should “partition” be “parturition” on ll. 302-303?, l. 521, l. 584), and inconsistent spelling (e.g. “savanna” or “savannah”? “mega-herbivore” or “mega herbivore”?). I think the manuscript would also benefit from a thorough read-through for sentence clarity. For example, the phrase “and their feeding strategies adopted” on l. 189 does not add anything to the sentence. Or on l. 571, “the drinking of water” can simply be “drinking”.

Experimental design

The search terms and databases that the authors used are stated, but the rest of the methods are not detailed enough and it is unclear how they affected the selection of literature reviewed. For example, one of the criteria for inclusion was that the article was available to the authors. What does this mean? How many articles were not available? If there are a large number of articles in this category I would think this would really affect the patterns you can derive from the literature.

Additionally, because the stated aim of the study is to review elephant movement in relation to minerals, “movement” and “habitat selection” should be terms included in the search. Asian elephants are discussed throughout the paper but “Elephas” is not a search term and therefore presumably not systematically reviewed.

Validity of the findings

The findings seem to be reflective of the studies reviewed but overall do not reflect the stated aims of the review.

Additional comments

l. 28: “The world’s natural ecosystem” is vague. There is no one world natural ecosystem.
ll. 29-30: This should be written as a list.
ll. 38-40: This point does not seem relevant to your larger framing. What does elephant capacity to alter their landscapes have to do with their altered movement as a result of anthropogenic activity? If the argument is that elephants are keystone species and their use as example species is therefore relevant to a range of other species, this should be stated.
ll. 47-50: What were the results of the systematic search after inclusion and exclusion criteria were applied? How did number of studies break down by year, geographical region, topic, etc…? What were the trends that you found in your search? What areas of research were glaringly missing from your search?
ll. 52-54: This sentence is redundant, as you have already stated the idea earlier in the abstract.
ll. 54-56: This review does not provide inference on predicting or mitigating human-elephant conflict.
ll. 56-57: This was already mentioned in the methods of the abstract and does not belong in the discussion.
l. 58: Generally, the introduction should start with broad context to set up the research question. As written it is a bit disjointed, with the stated aim of the paper expressed before the need for it is established. Reworking the introduction could focus on the general issue of access to resources across species on human-dominated landscapes, then narrow to elephants as a valuable case study, and finally lay out plans for the research question.
ll. 59-70: The abstract sets this up as a larger issue of wildlife populations being squeezed into smaller areas, and I think the introduction should be similarly structured. Elephants can then be brought in later in the introduction as a relevant case study.
l. 59: The paper discusses all three extant elephant species, so the emphasis should not only be Loxodonta africana but elephants generally if this paper is to retain discussion of the three species.
ll. 62-64: This sentence needs references.
ll. 66-67: This sentence is unrelated to the rest of the paragraph.
ll. 74-76: This statement needs a reference.
ll. 75-76: Again, I am unclear as to what the “world’s natural ecosystem” is.
ll. 77-80: Needs a reference.
ll. 80-82: Animals selecting habitat to meet their dietary needs is not something they adapt to do on human landscapes, it is something they have evolved to do generally. Perhaps what you are trying to say is that the way they go about this on human-dominated landscapes can present managers with new problems? Please clarify.
ll. 87-97: These objectives are not consistent with what is discussed in the rest of the paper.
ll. 101-103: This is an awkward sentence with which to end an introduction, and should probably be mentioned earlier to set up the need for your study.
ll. 129-134: Much more detail is needed here. How many studies were included in the final selection? How did the studies break down by topic, region, species, year? What were the trends in topics covered? Figures and tables would be very helpful here.
ll. 138-142: These aims are very different from those laid out in the introduction.
l. 143: Is this the introduction or discussion? Is it labeled incorrectly?
l. 143: It would be helpful for the first paragraph of the discussion (at minimum) to highlight the general trends found in the literature search. At present, the discussion reads as a collection of discrete and unrelated sections. A reworking of the discussion with a more logical organization would be helpful for readers.
l. 144: If both African species are a part of this review, forest elephants should be introduced earlier in the manuscript and savannah elephants should not be the sole focus in the title.
ll. 144-186: The relationship between activity budgets and your stated aims is unclear.
ll. 259-261: Again, if Asian elephants are included in this review they should be included systematically (i.e. in the search terms).
l. 264: This comes out of left field. What do equids have to do with this?
ll. 265-282: It would help to have some context for this section set up in the introduction, as it comes out of left field and is given considerable space in the discussion, which is a bit jarring. But more importantly, it is unclear what this section has to do with elephant movement related to minerals, or their movement on human landscapes.
ll. 268-270: Add reference.
l. 285-286: Add reference.
l. 286: What does this mean? African elephants grow tusks throughout their lives.
ll. 310-311: Add reference.
ll. 363-365: Rework this sentence for clarity.
ll. 373-402: The relationship between this paragraph and the larger idea of the paper is unclear.
l. 412, 421: What is Cn?
l. 415: What is Cot?
ll. 432-435: Add references.
ll. 500-502: Reference?
ll. 509-514: What does this have to do with movements for minerals?
ll. 542-544: Was this just speculation? What was the evidence they presented for this?
ll. 564-566: This seems very speculative.
ll. 592-593: Odd sentence.
ll. 594-596: Reference?
ll. 633-637: References?
ll. 668-670: This sentence is unrelated to the rest of the paragraph.
ll. 672-673: This claim is untrue. The reviewed work does not include elephant movement or habitat selection.
ll. 686-689: References?
Table 2: Kenya is not southern Africa.

Reviewer 3 ·

Basic reporting

The manuscript was well written and clear.
This review is of interest to a broad audience, as well as providing an interdisciplinary perspective for the reasons behind elephant and other herbivore movements. To my knowledge, this field has not previously been reviewed and the information is important to conservationists, ecologists, and animal managers.

Experimental design

The organization of the beginning of this article was slightly confusing. The authors had subsections for the abstract and then again went through those subsections before digging into the information. I would suggest removing the small discussion section as it’s repetitive and renaming the Line 143 as Feeding Behavior rather than “Introduction” again.
The survey methodology seems mostly comprehensive, the only concern I have is that it was stated in Line 114 that the authors reviewed articles only that were available to them in full. I wonder what the restrictions were in their access to journals and how much was not reviewed due to lack of access. I am also surprised that "movement" was not included as a search term given the title and focus of this article. Although even with these slight concerns, it seems like significant literature was covered.

Validity of the findings

The literature reviewed supports the goals set out by the authors and they conclude with valid gaps and future directions.

Additional comments

This is a valuable review of the literature! Just a few edits and comments below:
Line 176 add “to” before adjust
Line 185 I think this was supposed to be dry season given the preceding text.
Line 212-213 Introduce the abbreviations in the previous sentence and you may want to quickly define “as-fed”
Line 343 I think if you define this abbreviation earlier in the text you won’t have to re-establish it here
Line 412 Define Cn or is this supposed to be Cu?
Line 421 is this supposed to be Cn?
Line 428 Need to define P as abbreviation for phosphorus

---

## Round 0.2 · Major Revisions

Thank you for resubmitting your article for review to PeerJ. Two of the three experts who reviewed your original submission have reviewed your revised version of your article. While both note that you have made considerable improvements to your article, both still have some concerns with your review and request additional edits. Please respond to their comments in detail as you prepare your article for re-submission.

In addition to the two reviewers' feedback, I also have some of my own comments that I wish to make.

Firstly, I wish to flag reviewer 3's concern about the structure of your review because I agree with them that the order in which you present your information is confusing.

The current order of your subsections is this:
1. Introduction
2. Survey Methodology
3. Databases searched
4. Fields searched
5. Grey literature reviewed
6. Results
7. African savanna elephant feeding behaviour
8. Human-elephant conflict (HEC)
9. Challenges of estimating elephant nutritional requirements
10. Reported mineral deficiencies in captive and free-living elephants
11. Environmental geochemistry as a driver for elephant dietary intake
12. Geochemistry influence on herbivore animal movement
13. Movement choices of herbivores
14. Movement choices of elephants
15. Nutritional factors affecting elephant movement
16. Conclusions

It is unclear to me how this order relates to the order of your stated aims. Additionally, I do not understand why you have disrupted your discussion of elephant nutritional needs with a section on HEC. I believe this section would be better placed at the end as an application of your review. I am also unsure why, given the focus of your review was on elephants, you include a subsection on herbivores more generally. I suggest either deleting this, or moving this section to the end as another application of your review.

Personally, I think the order of your subsections should be:
1. Introduction
2. Methods
3. Results
4. Elephant nutritional needs
5. Elephant feeding behavior (i.e. how they meet the above identified needs)
6. Elephant movement patterns, as related to geochemistry/nutritional factors
7. Applications to ameliorating HEC
8. Applications to other herbivore species in comparable environments

Secondly, I still found the reporting of your methods and results to be unclear. I have provided specific comments in an annotated copy of your manuscript. Please find it attached. Reviewer 1 has also appended an annotated copy of your review with comments and suggested edits.

·

Basic reporting

Significantly improved. See attached file.

Experimental design

Clear structure, revisions made. See attached file.

Validity of the findings

Clarified. See attached file.

Additional comments

In my opinion the manuscript was significantly improved and there is a clear structure in it now.
Please find my comments in the PDF attached.

Reviewer 3 ·

Basic reporting

The authors made changes to the introduction that clarified audience and motivation.

Experimental design

The authors made suggested improvements to clarify their methods and their citations. After the first revisions, I still think the organization of the review is unclear. I think to make it easier to follow, the authors should use a consistent presentation of the ideas following what they introduce as numbered list of objectives in the introduction and in lines 103-108. So they should either change those lists to follow how their paragraphs are presented or vice-versa. I also think the HEC paragraph that was added should either be relocated to a different part of the manuscript or should be better connected to the preceding paragraph. I currently don't see how it follows from the mineral needs of elephants. Do the authors want to introduce it as a type of feeding behavior? Or as a result of nutritional requirements?

Validity of the findings

The authors have presented evidence that supports their objectives and future directions are identified.

---

## Round 0.3 · Minor Revisions

Thank you very much for responding to mine and the reviewers' comments and suggestions. I believe that your review is now much clearer and is suitable for publication in PeerJ. Congratulations!

However, I did spot a few typos throughout. As PeerJ does not offer proofing, I took the liberty of highlighting some typos that I noticed as I reviewed your article. Please find them highlighted in the attached file. Please correct these and also check your article thoroughly for other such errors before resubmitting. Once you have attended to these, it will be my pleasure to accept your article for publication in PeerJ.

---

## Round 0.4 · accepted · Accept

Thank you very much for attending to my final few requests. It is my absolute pleasure to accept your article for publication in PeerJ. I believe that your literature review will provide a useful resource for many in the field.

#